# First Detection and Molecular Characterization of *Pseudomonas aeruginosa bla*_NDM-1_ ST308 in Greece

**DOI:** 10.3390/microorganisms11092159

**Published:** 2023-08-26

**Authors:** Katerina Tsilipounidaki, Christos-George Gkountinoudis, Zoi Florou, George C. Fthenakis, Vivi Miriagou, Efthymia Petinaki

**Affiliations:** 1Faculty of Medicine, University of Thessaly, 41500 Larissa, Greece; tsilipoukat@gmail.com (K.T.); gountinoudis@gmail.com (C.-G.G.); zoi_fl@yahoo.gr (Z.F.); 2Veterinary Faculty, University of Thessaly, 43100 Karditsa, Greece; gcf@vet.uth.gr; 3Laboratory of Bacteriology, Hellenic Pasteur Institute, 11521 Athens, Greece; miriagou@pasteur.gr

**Keywords:** antibiotic resistance, *bla*
_NDM-1_, carbapenem, Greece, *Pseudomonas aeruginosa*

## Abstract

The objective of the present study is to report the detection and the molecular characterization of nine *bla*_NDM-1_-positive *Pseudomonas aeruginosa* isolates, all of which belonged to the epidemic high-risk international clone ST308, and all were isolated from patients in a tertiary care hospital in Central Greece from May to July 2023.The isolates were characterized by whole genome sequencing to obtain multi-locus sequencing typing (MLST) and identify the *bla*_NDM1_-environment and resistome and virulence genes content. In silico MLST analysis showed that all isolates belonged to the high-risk ST308 international clone. All strains possessed 22 different genes, encoding resistance to various antimicrobial agents. Whole genome sequencing revealed that the *bla*_NDM-1_ was chromosomally located within the integrative and conjugative element ICE_Tn*4371*_*6385* and that it was part of one cassette along with two other resistance genes, *floR* and *msrE*. Two additional resistance cassettes were also found in the genome, which included the arrays of *aph(6)-Id*, *aph(3″)-Ib*, *floR*, *sul2* and *aadA10*, *qnrVC1*, *aac(3)-Id*, *dfrB5*, *aac(6′)-II*. Additionally, the strains possessed various virulence genes, e.g., *aprA*, *exoU*, *lasA*, *lasB*, *toxA*, and *estA*. All of the isolates shared identical genomes, which showed 98% similarity with the *P. aeruginosa* ST308 genome (acc. no CP020703), previously reported from Singapore. To our knowledge, this is the first report of ST308 *bla*_NDM-1_-positive *P. aeruginosa* isolation in Europe, which indicates the transmission dynamics of this high-risk clone.

## 1. Introduction

*Pseudomonas aeruginosa* has become an important cause of Gram-negative infections worldwide, especially in immunocompromised patients [1,2]. It causes infections in wounds (especially in burn patients), the urinary tract, bloodstream, surgical sites, eye, external ear, and the respiratory tract. As a frequent causal agent of hospital-acquired infections (HAI), it is the most common pathogen isolated from patients who have been hospitalized for over one week [3]. In recent years, nosocomial infections caused by *P. aeruginosa* have been recognized as a significant problem due to the intrinsic resistance of the organism to many classes of antibiotics and its capacity to acquire practical resistance to all effective clinically available antibiotics [4]. This has led the World Health Organization to categorize *P. aeruginosa* in the first list of ESKAPE pathogens (*Enterococcus faecium*, *Staphylococcus aureus*, *Klebsiella pneumoniae*, *Acinetobacter baumannii*, *Pseudomonas aeruginosa*, and *Enterobacter* species) as a top priority (critical) organism for research, discovery, and new drug development (https://www.who.int/news/item/27-02-2017-who-publishes-list-of-bacteria-for-which-new-antibiotics-are-urgently-needed; accessed on 25 June 2023).

In *P. aeruginosa*, carbapenem resistance stems from a combination of *β*-lactamases, porin mutations, MexA-MexB-OprM efflux pump overexpression, and/or penicillin-binding protein alterations [5,6]. All types of transferable carbapenemases, except SIM-1 (Seoul imipenemase), have been detected in *P. aeruginosa* isolates around the world. Among them, the metallo-beta-lactamases (MBLs) are considered to be the most clinically important. Most genes encoding MBLs have been found as gene cassettes in integrons and are transferable. Further, more resistance genes for other classes of antibiotics can be present in the same integrons, thus contributing to the development of multi-drug-resistant phenotypes [6].

New Delhi metallo beta lactamase-1 (NDM-1) is a MBL that confers resistance to all *β*-lactam antibiotics and beta-lactam/ inhibitor combinations such as ceftazidime/avibactam, with the exception of aztreonam [7]. However, many strains that harbour *bla*_NDM-1_ are also aztreonam-resistant, presumably due to a different resistance mechanism. The *bla*_NDM-1_ gene is located on mobile genetic elements, harbouring multiple resistant determinants, thereby conferring extensive drug resistance, leaving only a few or no therapeutic options [8]. NDM-1 has been identified mostly in *Escherichia coli*, *Klebsiella pneumoniae*, and, to a lesser extent, *Acinetobacter* and *Pseudomonas*. *P. aeruginosa* strain carriers of the *bla*_NDM-1_-encoding gene have been found to circulate in various countries internationally, belonging in different Sequence Types (STs): ST773 in Libya; STs 38, 773, 235, 357, and 654 in Asia; STs 235 and 654 in Serbia; ST 773 in Korea; ST654 in Chile; ST274 in Ethiopia; ST1966 in China; and ST 308 in Singapore and Malaysia [9,10,11,12,13,14,15,16,17].

In Greece, the rate of carbapenem-resistant isolates of *P. aeruginosa* varies between hospitals and, in general, it is around 35% of the total isolates (http://www.mednet.gr/whonet/; accessed on 25 May 2023). Moreover, previous studies have demonstrated that the production of *bla*_VIM_ is the most common mechanism in carbapenem-resistant isolates of *P. aeruginosa* recovered in Greece [18].

The objective of the present work is to report the detection and the molecular characterization of nine *bla*_NDM-1_-positive *P. aeruginosa* isolates isolated from patients in a tertiary care hospital in Central Greece from May to July 2023. All the strains belonged to the ST308 clone, an epidemic high risk clone that succeeded worldwide in the context of hospital infections.

## 2. Materials and Methods

### 2.1. Isolation of bla_NDM-1_ P. aeruginosa

From May to July 2023, nine *bla*_NDM-1_ *P. aeruginosa* isolates were recovered from different patients who were hospitalized at the University Hospital of the University of Thessaly, located in Larissa. The hospital is a tertiary care hospital in Central Greece. The identification and the susceptibility testing of the microorganisms were carried out using the automated Vitek-2 system (BioMerieux, Marcy l’ Etoile, France) [18]. The minimal Inhibitory Concentration (MICs) values of imipenem and meropenem were determined by using the MIC test strip (Lofilchem, Roseto degli Abruzzi, Italy); MIC to colistin was determined by using the broth microdilution method, following the respective EUCAST guidelines (www.eucast.org). As the isolates were found to be phenotypically resistant to imipenem (with MIC > 4 mg L^−1^), they were tested for the presence of carbapenemase-encoding genes such as *bla*_VIM_, *bla*_IMP_, and *bla*_NDM_ via a relevant PCR followed by sequencing analysis [19]. Subsequently, all isolates were further characterized by whole genome sequencing (WGS).

### 2.2. Whole Genome Sequencing of bla_NDM-1_-Positive P. aeruginosa

Initially, the libraries were prepared using Ion Torrent technology and Ion Chef Work flows (Thermo Fisher Scientific, Waltham, MA, USA). Genomic DNA libraries were sequenced on the S5XLS system, and the analysis of primary data was conducted using Ion Torrent Suite v.5.10.0 (Thermo Fisher Scientific). The quality of the reads was checked using FastQC software version 0.11.9. The reads for each sample were assembled using the SPAdes genome assembler v3.15.5 with the default parameters. The quality of the assembled genomes was assessed with the Quast version 5.2.0 tool. The average coverage for each genome was computed using the mapPacBio tool from BBTools (https://sourceforge.net/projects/bbmap/; accessed on 10 July 2023). Gaps were filled by the sequencing of the overlapping PCR-produced fragments.

The typing of isolates was assessed by using the online tool MLST 2.0 [19]. The identification of genes coding for antibiotic resistance in the assembled genomes was performed by using the online tool ResFinder-4.1, with the ID threshold set to 90% and the minimum length set to 60%. The characterization of virulence factors was performed by using the Virulence Factor Data Base. In order to determine the genetic contexts of *bla*_NDM-1_-encoding genes, a BLAST analysis was performed. Only results with a high identity score (100% identity and ≥90% cover age) were considered. The genomes of our strains were compared using Blast analysis with the international *bla*_NDM-1_ ST308 genome (acc. no CP020703), as previously reported [16,20].

### 2.3. Nucleotide Accession Numbers

The genomes of the first two *bla_N_*_DM-1_-positive *P. aeruginosa* isolates (A3453, A3454) have been deposited in GenBank under BioProject accession PRJNA993831 (Assemblies GCA_030504675.1 and GCA_030504695.1, respectively).

## 3. Results

### 3.1. Characteristics of Patients

Eight of the patients from whom the *bla_N_*_DM-1_-positive *P. aeruginosa* were isolated had been hospitalized in the Intensive Care Unit of the University Hospital of Thessaly. The ninth strain was recovered from a patient admitted into the hospital for orthopaedic surgery. All patients had been hospitalized prior to the isolation of these strains for more than five days.

The first strain (A3453) was isolated on 16 May 2023 from bronchial secretions of a female patient; the patient had previously been hospitalized in the ICUs of various hospitals in Greece prior to admission at the University Hospital of Thessaly in April 2023. The last strain (A3542) was recovered on 10 July. Details for patients and clinical specimens are shown in Table 1.

### 3.2. Antimicrobial Susceptibility Profiles of bla_NDM-1_ Pseudomonas aeruginosa

During the two-month period from 11 May to 10 July, a total of 42 carbapenem-resistant isolates of *P. aeruginosa* were recovered at the University Hospital of Thessaly. Of those, nine (21.4%) were found to be *bla*_NDM-1_-positive.

These nine isolates shared the same resistance phenotypes to amikacin (MIC > 64 mg L^−1^), aztreonam (MIC > 64 mg L^−1^), cefepime (MIC > 64 mg L^−1^), ceftazidime (>64 mg L^−1^), ciprofloxacin (>4 mg L^−1^), imipenem (>16 mg L^−1^), levofloxacin (>8 mg L^−1^), meropenem (>16 mg L^−1^), piperacillin (>128 mg L^−1^), piperacillin plus tazobactam (>128 mg L^−1^), ticarcillin (>128 mg L^−1^), ticarcillin plus clavulanic acid (>128 mg L^−1^), and tobramycin (MIC > 16 mg L^−1^), but remained susceptible to colistin (MIC = 1 mg L^−1^).

### 3.3. Multi-Locus Sequence Typing (MLST)

In silico performed MLST revealed that all isolates belonged to the international ST308 clone [16,20].

### 3.4. Identification of Resistance and Virulence Genes

The WGS analysis revealed that each of the nine *bla*_NDM-1_-positive *P. aeruginosa* strains possessed 22 distinct resistance genes. Specifically, each of the strains carried four genes associated with resistance to *β*-lactams (*bla*_NDM-1_, *bla*_PAO_, *bla*_OXA-10_, *bla*_OXA-488_), seven genes associated with resistance to aminoglycosides (*aph(3′)-Ib*, *aph(6)-Id, aac(3)-Id*, *aac(6′)-Ib3*, *aac(6′)-II*, *aadA10*, *rmtF*), three genes associated with resistance to quinolones (*crpP*, *aac(6′)-Ib-cr*, *qnrVC1*), three genes associated with resistance to folate (*sul1*, *sul2*, *dfrB5*), two genes associated with resistance to chloramphenicol (*catB7*, *floR*), and one gene associated with resistance to macrolides (*msrE*), fosfomycin (*fosA*), and quaternary ammonium (*qacE*).

A sequence comparison analysis revealed that the *bla*_NDM-1_ genetic environment was identical to that previously identified in *P. aeruginosa* strain ST308, which was first isolated in Singapore [16,21].

The *bla*_NDM-1_ was chromosomally inserted within an integrative and conjugative element (ICE) ICE_Tn*4371*_*6385* and was part of one cassette along with two other resistance genes, *floR* and *msrE* (Figure 1). This region of 74.2 kb was also punctuated by three IS*91* elements.

Moreover, two other cassettes with antimicrobial resistance genes were found in the genome, which were identical to those described previously [16,21]. The first cassette included *aph(6)-Id*, *aph(3″)-Ib*, *floR*, and *sul2*, and the second included *aadA10*, *qnrVC1*, *aac(3)-Id*, *dfrB5*, *aac(6′)-II* [16,21]. Whole genome alignment revealed two other genomic regions that were integrated in the chromosome: the Dobby phage cassette and the WYL domain-containing protein cassette that did not contain antibiotic resistance genes [16,21].

The isolates also possessed various virulence genes, including *aprA*, *exoU*, *lasA*, *lasB*, *toxA*, *estA*, *pcrD*, *spcS*, *rhlA*, *rhlB alg8*, *algB*, *algC*, *algE*, *algG*, *algQ*, *algR*, *algU*, *algW*, *mucA*, *mucB*, *mucD*, *mucP*, *cupA1*, *cupA2*, *cupA3*, *cupA4*, *cupA5*, *cupB1*, *cupB2*, *cupB3*, *cupB4*, *cupB5*, *cupB6*, *cupC3*, *hdtS*, etc.

All the nine isolates were identical, which showed 98% similarity with the genome CP020703 [16].

## 4. Discussion

High-risk *P. aeruginosa* ST308 has been reported to carry carbapenemase genes such as *bla*_VIM_ and *bla*_IMP_, but thus far, it has been rarely associated with *bla*_NDM-1_ [22,23]. The circulation of strains ST308, carriers of *bla*_NDM-1_, has been reported only in Singapore and Malaysia [16,17,24]. To our knowledge, in Europe, *bla*_NDM-1_-negative ST308 strains circulate in Germany, France, and Spain, whereas such clones have not previously been identified in Greece [22,25,26].

The first ST308 carrier of the *bla*_NDM-1_ strain was detected in our hospital in May 2023, and subsequently, until now (10 July), another eight strains were also isolated. All the strains shared the same genetic background and carried three cassettes with antimicrobial resistance genes, the first with *aph(6)-Id*, *aph(3″)-Ib*, *floR*, and *sul2*, the second with *aadA*, *qnrVC1*, *aac(3)-Id*, *dfrB5*, and *aac(6′)-II*; and the third with *bla*_NDM-1_, *msr*(E), and *floR*. This third cassette was located in ICE_Tn*4371*_*6385* [16,21].

Given that the strains isolated in Greece had a very high similarity with the strains located in South East Asia, a query regarding when and how this clone was first introduced in Greece arises. The index patient was a young woman who was hospitalized in various Greek ICUs prior to admission in our hospital. No surveillance screening for the carriage of carbapenem-resistant isolates of *P. aeruginosa* was performed during initial admission in the hospital; one cannot be certain whether the strain was introduced by her or whether it was already circulating in the hospital.

From spring 2020 (i.e., at the emergence of COVID-19) to spring 2023, the national policy for the national health system was to give significant priority to the detection of COVID-19 in order to control the pandemic. Hence, the majority of relevant resources had been directed towards that objective; in turn, this resulted in a decrease in other monitoring activities (including surveillance for multi-resistant pathogens) within hospitals. Hence, the exact time and route of ST308 *bla*_NDM-1_ transmission within the hospital are unclear. As the situation with COVID-19 has been progressively relaxing and given the isolation of this clone, a strict surveillance plan has been initiated in the hospital; this includes patient carriage cultures and environmental samples cultures, in order to monitor and limit the spread of this clone.

The impact and the consequences of the first isolation and emergence of ST308 *bla*_NDM-1_ *P. aeruginosa* in Greece is two-fold. First, the therapeutic protocols against infections by such strains will need to be based on a combination of ceftazidime/avibactam and aztreonam, as it is evident that these strains possess multiple resistance mechanisms additional to *bla*_NDM-1_ [27]. The administration of cefiderocol could be a further therapeutic option for these strains, but recently, the emergence of isolates resistant against it as well has been described [28]. Second, *P. aeruginosa* ST308 has been identified as a major contaminant in the hospital water plumbing networks, where it was identified as a copper tolerant clone due to the presence of genomic island number 7 [29]. This clone showed intraclonal diversity as it can adapt to various environments and could acquire genes such as those on mobile genetic elements, thus leading to mutations in its genome [30].

## Figures and Tables

**Figure 1 microorganisms-11-02159-f001:**
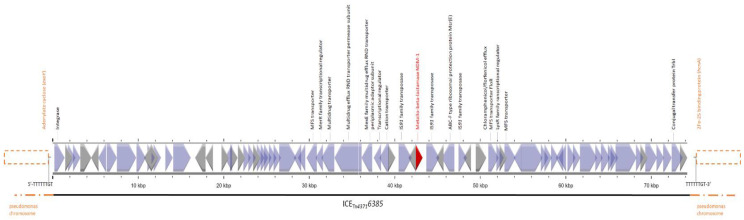
Graphical presentation of ICE_Tn*4371*_*6385* detected in our *bla*_NDM-1_-positive *Pseudomonas aeruginosa* strains, as found during whole genome sequencing.

**Table 1 microorganisms-11-02159-t001:** Details regarding the isolation of nine *bla_N_*_DM-1_-positive *P. aeruginosa* strains at the University Hospital of Thessaly from May to July 2023.

Isolate Reference	Patient Age/Sex	Previous Hospitalization	Recent Travel Abroad	Date of Isolation	Hospital Department	Type of Clinical Specimen
A3453	55/woman	Yes/multiple ^1^	No	16 May 2023	Intensive Care Unit	Bronchial secretions
A3454	81/woman	Yes/LTCF ^2^	No	19 May 2023	Intensive Care Unit	Central venouscatheter
A3462	68/man	No	No	25 May 2023	Intensive Care Unit	Blood
A3468	65/woman	No	No	30 May 2023	Intensive Care Unit	Bronchial secretions
A3472	36/man	Yes/LTCF	No	31 May 2023	Intensive Care Unit	Central venouscatheter
A3504	49/man	Ye s/LTCF	No	18 June 2023	Intensive Care Unit	Blood
A3529	66/man	No	No	04 July 2023	Intensive Care Unit	Urine
A3534	27/man	No	No	06 July 2023	Intensive Care Unit	Urine
A3542	79/woman	No	No	10 July 2023	Orthopaedic Ward	Tissue

^1^ admissions in various Greek hospitals; ^2^ LTCF: Long Term Care Facility.

## Data Availability

All the data associated with this manuscript are provided within the manuscript.

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
