# Peer review of "First Detection and Molecular Characterization of Pseudomonas aeruginosa blaNDM-1 ST308 in Greece"

_microorganisms, 2023, doi:10.3390/microorganisms11092159_

Round 1

Reviewer 1 Report

Interesting short communication by Tsilipounidaki et al on the isolation and characterisation of Pseudomonas aeruginosa blaNDM-1 ST308 in Greece from two patients attending a local hospital in the country.

Please find comments below comments to make this manuscript a better scientific read.

-The topic is quite misleading. It should not be an outbreak since there was no surveillance within a timeframe prior to your findings due to Covid and the index patients was admitted in various other hospitals previously as stated in your manuscript on lines 179-190.  Therefore, it would be inconclusive to say this is an outbreak. Please edit topic accordingly or provide patient history or surveillance report for Greece hospitals for the time gaps.

Line 12 – characterized instead of studied.

Line 29: is this figure for Greece alone? If not, what is the implication of this pathogen in the global perspective.

Line 51: it is not clear why this date was targeted. Any justification for this date?

Line 53: P. aeruginosa. Same for line 66

Provide references for methods on lines 56-62, explain how they were carried out, if/no modifications were made or were carried out strictly according to manufacturer instructions.

Line 108: it would be helpful to have a brief introduction of this gene in the introduction to contextualise the importance.

Line 116-117: Please state how this finding relates to other strains locally and internationally? Need to give a background to this clone in the introduction.

Not applicable 

Author Response

Interesting short communication by Tsilipounidaki et al on the isolation and characterisation of Pseudomonas aeruginosa blaNDM-1 ST308 in Greece from two patients attending a local hospital in the country.Please find comments below comments to make this manuscript a better scientific read.                  

We thank the reviewer for the supporting comments.

The topic is quite misleading. It should not be an outbreak since there was no surveillance within a timeframe prior to your findings due to Covid and the index patients was admitted in various other hospitals previously as stated in your manuscript on lines 179-190. Therefore, it would be inconclusive to say this is an outbreak. Please edit topic accordingly or provide patient history or surveillance report for Greece hospitals for the time gaps.                  

The term ‘outbreak’ has been deleted in the revised manuscript and appropriate changes were made in the title and the text, as suggested. The revised manuscript focuses on the description for the first time of a ST308 blaNDM-1 P. aeruginosa in Greece (and likely also in Europe).

Line 12 – characterized instead of studied.                  

The sentence was corrected

Line 29: is this figure for Greece alone? If not, what is the implication of this pathogen in the global perspective.                 

  A new sentence was added concerning the implication of P. aeruginosa in a global perspective.

Line 51: it is not clear why this date was targeted.  Any justification for this date?                  

In accord with guidelines issued by the Greek Ministry of Health, all hospital laboratories have stopped the molecular detection of SARS-CoV-2 since April 2023, which allowed us to redirect resources. Hence, in May 2023, routine surveillance for multi-drug resistant pathogens, which had been temporarily stopped during the Covid-19 pandemic, has started again.

Line 53: P. aeruginosa. Same for line 66.                  

The sentence was corrected in both cases

.Provide references for methods on lines 56-62, explain how they were carried out, if/no modifications were made or were carried out strictly according to manufacturer instructions.                  

Relevant references were added.

Line 108: it would be helpful to have a brief introduction of this gene in the introduction to contextualise the importance.                  

A paragraph was added, which focused on blaNDM-1 encoding gene

Line 116-117: Please state how this finding relates to other strains locally and internationally? Need to give a background to this clone in the introduction.                  

A sentence was added, which focused on the background of ST308   

Reviewer 2 Report

The main aim of the manuscript is clear which is “o report the detection and the molecular characterization of nine blaNDM-1 positive Pseudomonas aeruginosa isolates, isolated from patients in a tertiary care hospital in Central Greece during the period May to July 2023.”  it is relevant to microbiology and infectious diseases> It does not address a gap, but rather reveals the spread of some bacteria and the expansion of their geographical area to new areas according to the current study. It reveals the expansion and emergence of some new strains of certain bacteria in Europe for the first time, according to the results of the researchers.

Regarding the methodology: The author can add baseline patient characteristics data.

More references can be supportive

We think it is better to reinforce and improve the vocabulary of the manuscript  title

it is very interesting and well written manuscript. You have arisen some questions though, it would be great if you search for the source of entry such strains in europe. 

You can present more data  and results about these strains or even patients demographic data involved which may disclose some ambiguity of this transmission from continent to another one.

Author Response

The main aim of the manuscript is clear which is “o report the detection and the molecular characterization of nine blaNDM-1 positive Pseudomonas aeruginosa isolates, isolated from patients in a tertiary care hospital in Central Greece during the period May to July 2023.”  it is relevant to microbiology and infectious diseases> It does not address a gap, but rather reveals the spread of some bacteria and the expansion of their geographical area to new areas according to the current study. It reveals the expansion and emergence of some new strains of certain bacteria in Europe for the first time, according to the results of the researchers.                  

We thank the reviewer for the supporting comments.

Regarding the methodology: The author can add baseline patient characteristics data.                 

  In accord with suggestion, further details of patients, from which the pathogens were isolated has been added.

More references can be supportive                  

Further relevant references (13) have been added. We think it is better to reinforce and improve the vocabulary of the manuscript  title                   The title of the revised manuscript has been modified in accord with the comments and as suggested. Moreover, the word ‘outbreak’ is not cited anywhere in the text of the revised manuscript.i

t is very interesting and well written manuscript. You have arisen some questions though, it would be great if you search for the source of entry such strains in Europe.                   

To the best of our knowledge, no reports are available from European countries with regard to the appearance of ST308 blaNDM-1 P. aeruginosa strains. Unfortunately however, we cannot fully confirm the source of this clone in Greece (nor the period, when it first occurred in a hospital), because for the period spring 2020 to spring 2023, the national policy was to give significant priority to the detection of Covid-19, where the majority of our resources had been directed, and decrease all other monitoring activities. The routine surveillance for multi-drug resistant pathogens has restarted, in accord by national guidelines in April 2023.

You can present more data  and results about these strains or even patients demographic data involved which may disclose some ambiguity of this transmission from continent to another one.                  

In accord with suggestion, further details of patients, from which the pathogens were isolated has been added.

Author Response

This short communication reported the detection and the molecular characterization of nine blaNDM-1 positive Pseudomonas aeruginosa isolates, isolated from patients in a tertiary care hospital in Central Greece. This work is interesting and can help us better understand emerging bacterial isolates. Overall, this is an outstanding study with novel findings. However, I propose some suggestions that can improve the document. My overall recommendation as a reviewer to the editor is that the manuscript can be considered for publication after minor revision.                  

We thank the reviewer for the supporting comments.

1) It is highly suggested to revise this title and avoid the word “outbreak”.                  

The title of the revised manuscript has been modified in accord with the comments and as suggested. Moreover, the word ‘outbreak’ is not cited anywhere in the text of the revised manuscript.

2) The introductory section of this article lacks critical and in-depth analysis of the available literature and comprehensive relevant work.                  

We have added in the Introduction information about the distribution and significance of blaNDM-1 P. aeruginosa clones internationally, including the importance of the ST308.

For the methods of isolation of blaNDM-1 Pseudomonas aeruginosa, it would be good tocite and relate some other studies that investigated similar aspects.                  

Further references of other relevant studies have been added (references 9-15).

For antimicrobial susceptibility profiles, I highly recommend including sulfamethoxazole along with other antibiotics. Authors may refer to these below-mentioned articles in the methods section: https://doi.org/10.1016/j.biortech.2020.124403https://doi.org/10.1016/j.envpol.2021.116587; https://doi.org/10.1016/j.cej.2019.123674                   Unfortunately, EUCAST (the guidelines of which are followed in Greek hopitals) has not established breakpoints of this drug for P. aeruginosa isolates and does not include this drug in the routinely used drugs again this pathogen.

The quality of Figure 1 is very poor.                  

A new figure has been inserted.

Specific statements about Covid-19 are vague. Please rephrase it.                   The relevant passage has been rephrased, as suggested.  

Reviewer 4 Report

Find below the necessary revisions:

- Authors need to include the importance of detecting "blaNDM-1 positive Pseudomonas aeruginosa" at the beginning of the abstract.

- On line 46 replace "[" with ")"

- Throughout the discussion, authors should include information about the impact of these resistant strains on hospitals. What are the difficulties encountered for treatment and the therapeutic regimen used.

Author Response

  • Authors need to include the importance of detecting "blaNDM-1 positive Pseudomonas aeruginosa" at the beginning of the abstract.               The sentence has been corrected as suggested.-
  • On line 46 replace "[" with ")"                   The correction was made.-
  • Throughout the discussion, authors should include information about the impact of these resistant strains on hospitals. What are the difficulties encountered for treatment and the therapeutic regimen used.                   A new paragraph was added in the discussion about the impact of this clone on the hospital environment, the difficulties for treatment and the therapeutic protocols employed.    

Round 2

Reviewer 1 Report

Authors have addressed my comments accordingly. 

Just minor edits to be done which should be addressed by the editorial team

Reviewer 4 Report

After modifications, the manuscript can be accepted for publication.